# Nanotechnology in Cancer Diagnosis and Treatment

**DOI:** 10.3390/pharmaceutics15031025

**Published:** 2023-03-22

**Authors:** Noor Alrushaid, Firdos Alam Khan, Ebtesam Abdullah Al-Suhaimi, Abdelhamid Elaissari

**Affiliations:** 1Department of Stem Cell Biology, Institute for Research and Medical Consultations, Imam Abdulrahman Bin Faisal University, P.O. Box 1982, Dammam 31441, Saudi Arabia; 2Univ. Lyon, University Claude Bernard Lyon-1, CNRS, ISA-UMR 5280, 69622 Lyon, France; 3Biology Department, College of Science, Institute of Research & Medical Consultations (IRMC), Imam Abdulrahman Bin Faisal University, P.O. Box 1982, Dammam 31441, Saudi Arabia

**Keywords:** nanoparticles, nano-enabled formulations, cancer diagnosis, cancer imaging, cancer screening

## Abstract

Traditional cancer diagnosis has been aided by the application of nanoparticles (NPs), which have made the process easier and faster. NPs possess exceptional properties such as a larger surface area, higher volume proportion, and better targeting capabilities. Additionally, their low toxic effect on healthy cells enhances their bioavailability and *t*-half by allowing them to functionally penetrate the fenestration of epithelium and tissues. These particles have attracted attention in multidisciplinary areas, making them the most promising materials in many biomedical applications, especially in the treatment and diagnosis of various diseases. Today, many drugs are presented or coated with nanoparticles for the direct targeting of tumors or diseased organs without harming normal tissues/cells. Many types of nanoparticles, such as metallic, magnetic, polymeric, metal oxide, quantum dots, graphene, fullerene, liposomes, carbon nanotubes, and dendrimers, have potential applications in cancer treatment and diagnosis. In many studies, nanoparticles have been reported to show intrinsic anticancer activity due to their antioxidant action and cause an inhibitory effect on the growth of tumors. Moreover, nanoparticles can facilitate the controlled release of drugs and increase drug release efficiency with fewer side effects. Nanomaterials such as microbubbles are used as molecular imaging agents for ultrasound imaging. This review discusses the various types of nanoparticles that are commonly used in cancer diagnosis and treatment.

## 1. Introduction

The size of nanoparticles ranges from 10 nm to 100 nm and they can have a very large surface area, making them an attractive material for biological applications. The nanomaterials can easily move inside the body from one organ to another and can effectively penetrate the targeted tissues. The nanoparticles can be conjugated with drug molecules to target diseased tissues such as cancer cells for diagnostic purposes. Nanoparticles are smaller than blood cells and almost the same size as DNA [1]. This helps them to achieve better performance and gives them special physical, chemical, and optical properties that allow them to be used in the medical field to treat and diagnose cancer. In addition to being used to develop novel approaches, they can be used to enhance conventional methods. Moreover, nanoparticles can function as targeting agents to target specific molecules in cancer cells for better cancer imaging, which can improve cancer diagnosis. Several studies have demonstrated the potential of nanoparticles in enhancing cancer imaging for improved detection [2,3]. Nanomaterials are produced for precise applications in cancer diagnosis and detection and the main objective is to synthesize nanomaterials that are effective and can be easily delivered to cancerous tumor cells [4].

Based on their various biological applications, several types of nanoparticles, such as magnetic, carbon nanotubes, polymeric micelles, and liposomes, have been applied in cancer detection and imaging [5]. Because of their unique properties, these nanomaterials have been used in various medical applications. The unique chemical, optical, magnetic, and chemical properties of nanomaterials allow for the synthesis of imaging probes with better contrast, amplified sensitivity, controlled biodistribution, and better spatial imaging in MRI, PET, SPECT, and ultrasound techniques [6]. Among the various nanomaterials, microbubbles are used as molecular imaging agents and for ultrasound imaging. One study used vascular endothelial growth factor receptor 2 and αvβ3-targeted microbubbles for molecular ultrasound imaging, demonstrating their precise binding to angiogenic tumor blood capillaries [7]. The early detection of tumors and cancers remains a challenge in clinical diagnosis. The development of advanced nanomaterials is a promising approach to early cancer detection, diagnosis, and imaging. Metal nanoclusters can also be applied in cancer diagnosis and treatment [8,9,10,11,12]. In addition, researchers are studying how nanoparticle-induced toxicity can be reduced, as well as how to produce safe and efficient nanoparticles for human consumption. This review highlights and discusses the most important nanoparticles that are applied in cancer diagnosis and screening, as well as the challenges and shortcomings of the clinical use of various nanoparticles. 

Nanoparticles provide several advantages over small-molecule drugs, including prolonged circulation time and enhanced delivery to targeted sites. Once an NP enters the body, it interacts with the host’s immune system and is engulfed by cells of the mononuclear phagocyte system. The interaction between nanoparticles and the immune cells can result in immune suppression or immune stimulation, which may enhance or reduce the treatment effects of nanoparticles. 

## 2. Conventional Methods of Cancer Diagnosis via Imaging Technology

Imaging is the first step in cancer diagnosis and therapy. It is used to better understand the cancer stage, as well as the size and location of the tumor so that a treatment plan can be started. In the 16th century, the microscope was the main method used to detect cancer or diseases in cells. The X-ray machine was invented in 1983, followed by the ultrasound machine in 1956. Between 1972 and 1977, CT and MRI machines were invented [10]. Thereafter, medical imaging was developed and there was an improvement in the development of lenses from 2D to 3D materials, which improved cancer diagnosis. Today’s technology can give us a clear picture of living cells deep inside the body. One researcher was able to observe cellular function deep inside the body using engineered cells that formed air-filled proteins and detect cancer cells through ultrasound imaging, providing insights into the disease in cancer cells [10]. Today, many doctors have used these diagnostic methods to target specific cancer genes and proteins. Additionally, many studies have demonstrated the efficacy of using radiographic imaging such as CT and DNA detection, which can guide investigations based on the type of mutation(s) examined, cancer risk factors, and the relative occurrence of each cancer category. Breast cancer, for example, can be detected early using radiographic imaging. In 2014, over 235,000 new cases of breast cancer were diagnosed and over 40,000 deaths were reported in the United States. In addition, about 224,000 people were diagnosed with lung cancer and about 160,000 of these patients died [4].

Imaging technology has been developed for the detection of various types of cancer including lung cancer using annual low-dose CT (LDCT) [11]. Magnetic resonance imaging (MRI) scanning involves the use of a long tube that holds a large and very strong magnet. To examine a patient, the patient lies on a table that slides into the tube and the machine rotates around the patient emitting a powerful magnetic field. The MRI scanner uses a powerful magnetic force and a spurt of radiofrequency waves to pick up signals from the centers of hydrogen atoms in the body. Thereafter, the machine converts these signals into an image [12]. MRI is one of the methods used to generate pictures of soft body tissues that has been developed to detect cancer cells, which are sometimes difficult to examine using other imaging techniques. A comparison of traditional and nano-based methods of cancer treatment is shown in Figure 1.

## 3. Significance of Nanoparticles in Cancer Diagnosis 

There are many types of nanoparticles, including metallic, magnetic, polymeric, metal oxide, quantum dots, graphene, fullerene, liposomes, carbon nanotubes, and dendrimers, which are used in breast, colon, and cervical cancer diagnosis. Additionally, they are important for some imaging functions. It has been found that nanoparticles remain in the blood circulation for a long period of time before reaching the target cells, where they traverse many biological walls such as cell membranes and interact with biological systems (Figure 2). Moreover, cancer-specific antibodies can be conjugated with nanoparticles for better cancer binding and detection. Many recent studies have demonstrated that nanoparticles and sensors have significant potential for increasing the sensitivity of tumor detection and improving cancer diagnosis [13]. It has been reported that the detection of methylation patterns and mutations has been used as a marker for cancer diagnosis. It has been suggested that more studies are required before using extracellular vesicles, circulating tumor cells, and cell-free RNA in clinical diagnosis [14]. Researchers have used fluorescent gold nanoclusters (GNCs) or superparamagnetic (Fe_3_O_4_/GNCs), (γ-Mercaptopropyl) trimethoxysilane (MPS) as a stabilizing agent. The prepared GNCs@MPS was conjugated on the surface of “Fe_3_O_4_@SiO_2_ nanoparticles” followed by the addition of “poly ethylene glycol dimethacrylate (PGD)” to form Fe_3_O_4_/GNCs nanoprobes. It has been found that (Fe_3_O_4_/GNCs/Aptamer) is capable of uptake in HL60 cancer cells [15]. Various metallic nanoparticles have shown better sensitivity in detecting tumor cells as they can be conjugated with cancer-specific antibodies and penetrate cancer cells more effectively. Studies have reported that nanocomposites containing more than one type of nanoparticle, such as gold, platinum, silver, copper, and cobalt, are better candidates for cancer detection and imaging. The nanocomposites help to penetrate breast and colon cancer cells with great accuracy and produce better signals, which help to diagnose cancer. The unique optical, magnetic, and chemical properties of materials at the nanoscale allow the creation of imaging probes with improved contrast enhancement, increased sensitivity, controlled biodistribution, better spatial and temporal information, multifunctionality, and multimodal imaging across MRI, PET, SPECT, and ultrasound. These features could ultimately translate to clinical advantages such as earlier detection, real-time assessment of disease progression, and personalized medicine [15].

## 4. Synthesis of Nanomaterials

Nanomaterials can be produced using either synthetic approaches or green technology. Synthetic methods include chemical vapor deposition, thermal decomposition, hydrothermal synthesis, solvothermal synthesis, pulsed laser ablation, templating, combustion, microwave synthesis, gas phase synthesis, and conventional sol-gel. Each technique has its own advantages and disadvantages.

## 5. Nanoparticle Function in Cancer Image Enhancement and Contrasting Agents

Today, most studies are investigating ways to improve cancer imaging. Based on the functions of nanoparticles in the diagnosis and treatment of cancer, including photosensitizers, they have a significant impact on theragnostic applications today and are considered extremely helpful for personalized nanomedicine-based treatments [14]. MRI is an important application in which magnetic nanoparticles are used. One of the most commonly used magnetic nanoparticles is IONPs, which are used for cell tracking and imaging of the liver and lymph nodes. Magnetic nanoparticles are typically used for very sensitive devices [13]. Recently, nanotheranostics have been used for cancer therapy and chemical imaging due to their multimodal imaging capabilities. Nanotheranostics is a promising approach for cancer therapy and imaging and involves the use of complex nanoparticles made from materials such as calcium phosphate, polymeric, gold, proteins, and bismuth [16]. NPs have also been investigated for their potential to improve imaging, screening, and the early diagnosis of CRC (colorectal cancer) patients [10]. Colonoscopy is the most widely used technique for CRC diagnosis but other techniques, such as CTC, MRI, and transrectal ultrasonography (TRU), have also been used in the diagnosis of CRC. The application of MRI has shown remarkable improvements in CRC diagnosis. However, it has been found that MRI is not very effective in the precise identification of small polyps. Other techniques, such as fluoro-deoxy-glucose-positron emission tomography (FDG-PET) have been applied, but this technique has shown false-positive results in CRC patients due to radiotherapy [17]. Metallic nanoparticles have been used to detect breast and colon cancer by conjugating cancer-specific antibodies with nanoparticles to detect cancer in blood or urine samples. Biomarkers are proteins that can be found on the cell membrane, inside the cell, or surrounding the cell. By targeting these proteins, biomarkers can be diagnosed using different imaging applications such as MRI and CT. Photoacoustic (PA) imaging is an emerging imaging application that can detect high-resolution cancer tissues with good optical contrast. Moreover, the Raman imaging technique has also become one of the most effective applications for detecting cancer cells (Figure 3). This imaging method allows for improved guidance of tumor resection, and the accuracy of the imaging can be confirmed through histological analysis [16].

One of the main therapeutic methods used for various types of cancer is chemotherapy despite its high toxicity and severe side effects. Several studies have been conducted to find specific carriers for this therapy [10]. The application of nanoparticles and magnetic microcapsules has shown improvements in the anticancer effects of chemotherapy [10], particularly for liver and brain cancers. The drug delivery system can be improved by using biodegradable polymers, pro-drugs, and macromolecular matrix techniques. The synthesis of nanomaterials with magnetic properties has shown the lowest cytotoxic activity and magnetic nanomaterials have shown diverse biomedical applications, particularly in cancer diagnosis and therapy [18].

Nowadays, extracellular vehicles (EVs) are well-known nanocarriers, which may partially address the issues associated with chemotherapy. It has been reported that EVs can easily pass through the endogenous membrane due to their nanosize and may remain in the blood circulation longer for intrinsic cell targeting. EVs have lower toxicity and immunogenicity than other drug delivery systems. Studies have shown that EV-based drug delivery is effective in cancer therapy, in both in vitro and in vivo antitumor and anticancer models. There are several advantages of EVs such as better loading and targeting efficiency, better circulation time, and fewer side effects [19]. The application of various nanoparticles in cancer diagnosis is listed in (Table 1) [16].

## 6. Major Advantages of Nanomaterials

Conventional methods of cancer diagnosis, such as surgery, MRI, and PET, are very expensive. Nanoparticles possess exceptional characteristics, such as a higher surface area, volume proportion, and better targeting capabilities, and their gentle surfaces enhance their bioavailability and t-half by functionally penetrating the fenestration of epithelium and tissues. These particles have attracted attention in multidisciplinary areas, making them a promising material for many biomedical applications, especially in the treatment and diagnosis of various diseases. Today, many drugs are presented or coated with nanoparticles for the direct targeting of tumors or diseased organs without harming normal tissues/cells [10,11]. There are many types of synthesized nanoparticles, such as metallic, polymeric, and metal oxide nanoparticles; quantum dots; graphene; fullerene; liposomes; carbon nanotubes; and dendrimers, which have potential applications in cancer treatment and diagnosis. In many studies, nanoparticles have shown intrinsic anticancer activity due to their antioxidant capabilities, causing an inhibitory effect on the growth of tumors [10,11,12]. 

## 7. Commonly Used Nanoparticles in Cancer Diagnosis 

The application of nanoparticles in clinical diagnostics is called nanodiagnostics [10]. Recently, nanotechnology has developed to improve clinical diagnostics due to its increased sensitivity and capability for early cancer detection. There are several kinds of nanomaterials used in cancer diagnosis such as quantum dots, polymeric nanoparticles, carbon nanotubes, and dendrimers. To enhance the cancer detection capabilities of nanoparticles, they can be conjugated with aptamers, carbohydrates, antibodies, peptides, and other small molecules that specifically target molecules to reach the intended site [20]. Studies have been conducted that used gold nanoparticles for cancer diagnosis. The use of Au-doped nanoparticles showed extraordinarily high luminescence intensity [21]. Additionally, fluorescence has been utilized to detect the cancer biomarker CEA through color visualization. The limit of detection (LOD) was determined to be 10.00 ng/mL through a unique red color change, whereas a LOD of 0.10 ng/mL was achieved through differentiation of fluorescence intensity. The viability of this method was examined in real clinical samples [22]. It has been reported that an immunoassay strategy offers a greater ability to control the location of immobilized antibodies and has the potential for the precise analysis of the liver cancer biomarker Hsp90α [23]. Furthermore, miR-377-3p and miR-381-3p were used as the diagnostic biomarkers for CRC [24].

Nanoparticles, such as gold, silver, silica, magnetics, and iron oxide are used in cancer diagnosis due to their short detection time and low cost. In addition, they have fewer side effects than chemical-based treatments and radiotherapy. Electrochemical biosensors are simple, low-cost, and efficient, and are an excellent method for cancer diagnosis [24]. The cancer detection capabilities of nanoparticles can be improved through their functionalization. For example, in the detection of breast adenocarcinoma cells, antibodies against cancer were conjugated with polyethylene glycol (PEG). This antibody–PEG complex was then linked to the surface of nanoparticles through a sulfur-containing group located at the distal end of the PEG linker [20].

### 7.1. Metallic Nanoparticles

There are four types of metallic nanoparticles with sizes ranging from 1 to 100 nm, including metallic nanoplatelets, metallic nanostructures, metallic nanoparticles, and metallic nanowires. Metallic nanoparticles have a tendency to aggregate and form large structures due to the high energy on their surfaces, which promotes coalescence through thermodynamic processes. There are two main strategies for attaining the stabilization of metallic nanoparticles in a liquid medium. The first strategy is static stabilization, which involves the formation of an electrical double-layer through the absorption of negatively charged ions onto the nanoparticles. This charged layer can repel individual nanoparticles from each other and, consequently, prevents accumulation. The second strategy is steric stabilization, which involves encapsulating the metallic nanoparticles with a polymer, surfactant, or ligand to prevent the long protruding chains from aggregating. 

#### 7.1.1. Platinum Nanoparticles 

Platinum nanoparticles are widely used in medicine. Studies have shown that these platinum nanoparticles (PtNPs) possess intrinsic anticancer activity because of their antioxidant capabilities, which result in an inhibitory effect on the growth of tumors. In addition, targeting ligands linked to functionalized metal PtNPs have improved tumor targeting and PtNPs have facilitated better drug release and improved drug delivery efficiency. However, some recent studies have reported the toxic effects of nanoplatinum due to nanoparticle size, as nanoparticles were observed to accumulate in major organs and cells [25]. The application of platinum-based nanoparticles in cancer treatments is shown in Table 2. 

#### 7.1.2. Gold Nanoparticles

Gold nanoparticles (AuNPs) have many unique properties that can be useful for many biomedical applications [32]. AuNPs are small molecules and possess many unique properties for imaging techniques. In cancer imaging techniques, gold nanoparticles offer longer circulation times in the bloodstream with better tumor targeting for better-quality diagnoses (Figure 4). Moreover, AuNPs can be used in a wide variety of applications such as nucleic acid delivery, drug delivery, photothermal ablation, and radiotherapy [32]. AuNPs can be synthesized in various sizes and shapes and have great flexibility [32]. In addition, AuNPs have shown poor cytotoxicity and better biocompatibility, making them promising candidates for clinical applications (Table 3).

Several AuNP-based diagnostic products have been approved by the FDA and are available in the market while several other formulations are currently under trial phases [32]. The application of gold nanoparticles in various types of cancer detection and diagnosis is shown in Table 4.

### 7.2. Magnetic Nanoparticles

Magnetic nanoparticles are small in size (10–50 nm), have high binding properties, and exhibit several desirable characteristics such as the presence of electrons, protons, holes, and negative and positive ions. Furthermore, they are non-toxic and biocompatible. They are typically used as a label bioconjugate for the determination of cancer biomarkers (see Table 5). Moreover, the limit of detection is very low and they only require a short assay time compared with other nanoparticles. This is because the interaction for detection occurs in solution instead of on the surface of an electrode, which increases the contact time. They also have high saturation magnetization, an important requirement because the particles’ movement in the blood is controlled by an external magnetic field to reach the particles adjacent to the target cells and tissues. After administration, the particles can easily pass through the organs and tissues due to their small size. They also remain stable in water with a pH of 7. They can be injected into the body and be used to detect tumors or other abnormalities. Additionally, they can be used to manipulate cells and deliver drugs to specific locations in the body. The magnetic properties of nanoparticles make them ideal for use in magnetic resonance imaging (MRI) and an attractive option for a variety of applications. They are relatively easy to produce and can be modified to suit specific needs. Additionally, they are safe and non-toxic, making them an ideal choice for medical and environmental applications.

Nanoparticles can be synthesized in a variety of ways. They can be produced by chemical methods such as the sol-gel process or physical methods such as spray pyrolysis and laser ablation. The method used will depend on the desired properties of the nanoparticles. The size, shape, and composition of the nanoparticles can also be controlled. This allows for the production of particles with specific properties such as increased magnetic strength or improved drug delivery. Additionally, the particles can be modified to interact with specific molecules or cells. In addition, functionalized magnetic nanoparticles have been used for alternating magnetic fields or near-infrared light-induced cancer treatment [38].

Iron oxide nanoparticles are useful for in vivo studies because they are easily degradable and have a specific cytotoxic mechanism for uncoated iron oxide. Iron oxide nanoparticles are mostly used in magnetic-based cancer therapy where magnetic spin is used to generate oxygen radicals to detect cancer. In addition, these nanoparticles can be remotely controlled by an external electromagnetic field and can induce local toxicity mediated by reactive oxygen species and reactive nitrogen species for tumor therapy. This kind of therapy generates fewer side effects in normal and healthy tissues. Iron oxide nanoparticles loaded with antitumor drugs have added advantages over conventional antitumor drugs due to their ability to be remotely controlled, as demonstrated in a mouse model of breast cancer [39,40]. Breast cancer miRNA-155 was reported to overexpress during the development of breast cancer and is considered a cancer biomarker. However, miRNA-155 is difficult to map due to the lack of sensitive techniques. To overcome this problem, a new protocol has been developed for the rapid generation of magnetic nanoprobes that are able to measure miRNA-155 [41]. In another study, LCDIO was measured in cancer tumors for a longer duration, and circulating tumor accumulation was observed in brain tumors, making it sufficient for detecting tumors using MRI. Furthermore, it has been reported that LCDIO was specifically localized in tumor cells, but it can also be engulfed by macrophages and endothelial cells in areas of active angiogenesis. Additionally, in vitro cell culture results showed that the uptake of LCDIO was significantly correlated with the growth of tumor cells [42].

**Table 5 pharmaceutics-15-01025-t005:** Application of magnetic nanoparticles in cancer detection and screening.

Type of Nanoparticle	Cancer Cells	Applications	Citations
Magnetic gold nanoparticles	Breast cancer checks	ELISA-based detection of breast cancer, specifically for HER2 breast cancer patients.	[43]
Magnetic nanoparticles	Liver cancer cells	Enhanced detection of liver cancer cells (in vitro)	[22]
Magnetic nanoparticles	Brain cancer cells	Magnetic nanoparticles as contrast agents in the diagnosis and treatment of cancer (in vivo)	[44]
Surface-modified magnetic nanoparticles	Colon cancer cells	For colon cancer cell theranostics (in vitro)	[45]
Superparamagnetic iron oxide nanoparticles	Pancreatic cancer cells	Pancreatic cancer diagnosis using MRI and potential for early diagnosis through targeted strategies	[46]

### 7.3. Polymeric-Based Nanoparticles

Polymeric nanoparticles (PNPs) are solid materials with sizes ranging from 1 to 1000 nm. PNPs have two structure types, nanocapsules and nanospheres, and they have better mechanical strength, superior electrical conductivity, and better optical and thermal properties. The active pharmaceutical ingredients, drug delivery vehicles, and advanced fluorophores can be combined to achieve superior brightness with better biodegradability and low toxicity. They can also be microencapsulated, which is the most effective type of drug delivery system. Polymeric nanoparticles are widely used in vaccines and tissue engineering. These nanoparticles have the ability to protect the drug molecules during drug delivery. Natural polymers have been used to deliver drugs such as anticancer drugs, antiviral agents, vitamins, antioxidants, antisense oligonucleotides, and plasmid DNA. PNPs can be synthesized using synthetic polymers and there are two types, biodegradable and non-biodegradable. Poly-acrylates are non-biodegradable polymers that are used in drug delivery through dermal and transdermal routes but to a lesser extent than biodegradable polymers. PLGA (poly-lactic-co-glycolic acid), which is a biodegradable polymer, has been used in drug delivery through transdermal routes [47]. PNPs have been used for better optical and magnetic resonance imaging for brain cancer diagnosis [48]. Dendrimers are a 3D core–shell polymer that can cross the blood–brain barrier for better targeting. Poly-amidoamine is one of the most widely used dendrimers for drug delivery and dendrimers conjugated with tamoxifen are used as drug carriers [48]. The synthesis of polymeric-based nanoparticles typically involves the use of a surfactant, which helps to reduce the surface tension of the solution. The surfactant is then mixed with the monomer, which is the building block of the polymer. The monomer is then polymerized, forming nanoparticles.

Polymeric-based nanoparticles have some disadvantages such as low stability and a tendency to aggregate. These properties can make them difficult to use in some applications such as drug delivery. Additionally, polymeric-based nanoparticles can be difficult to synthesize, as the reaction conditions must be carefully controlled. Polymeric-based nanoparticles are also prone to oxidation, which can limit their use in some applications. Finally, polymeric-based nanoparticles can be difficult to scale up for industrial applications, as the reaction conditions must be carefully monitored. Doxorubicin-loaded protease-activated near-infrared fluorescent polymeric nanoparticles have been used for imaging and therapy for cancer [49]. In another study, the biodistribution of 99mTc-PLA/PVA/Atezolizumab nanoparticles has been used for the diagnosis of non-small cell lung cancer [50].

### 7.4. Metal Oxide Nanoparticles

Metal oxide nanoparticles, including NiO, ZnO, MnO_2_, Fe_2_O_3_, TiO_2_, and Co_3_O_4_, are mixed-metal oxides that have recently been used in electro-analysis for the detection of biomolecules. The use of metal oxides in detecting biomolecules offers advantages such as better biocompatibility for enzymes, which can improve detection accuracy. Another feature of metal oxide nanoparticles is that they have the ability to change their structure, which can affect the conductivity and chemical properties of the nanoparticles [41]. In addition, transition metal oxides have the ability to degrade various dyes in the presence of sunlight or UV light irradiation. There is an increasing interest in using sustainable and green synthesis-based materials, including metal oxides, in biogenic processes. The green process has many benefits such as using low-cost bioconstituents derived from various plant sources, which can be easily scaled up to synthesize transition metal oxide nanostructures [42]. Metal oxide nanoparticles have many applications and are commonly used in therapeutic and diagnostic contexts. For example, zinc oxide (ZnO) nanoparticles have been used as a treatment for diabetes-related cardiovascular disease, where streptozotocin was injected into rats with diabetes. The results showed that a low dose of treatment of ZnO nanoparticles facilitated the protection of cardiac cells from injury by reducing serum cholesterol levels [43].

### 7.5. Quantum Dots

Quantum dots (QDs) are nanoscale crystals, which have the ability to transport electrons. Under UV light exposure, QDs can emit light of various colors with very high energy. These QDs have many applications, including in the development of nanocomposites, solar cells, and fluorescent biological labels. There have been improvements in the diagnosis of cancer using high-resolution cellular imaging. It has been found that the fluorescence properties of quantum dots change upon exposure to different chemicals. Additionally, QDs have a passive site on their surface, where specific antibodies can be easily conjugated. There have been several bio-applications of QDs in imaging. Moreover, QDs have been used in drug delivery and cancer therapy, such as treatment for lung cancer, and can help to kill bacteria-related infections. In addition, conjugated QDs have been shown to inhibit the P-glycoprotein gene and protein expression in lung cancer cells by inducing miR-185 and miR-34b. miR-185 and miR-34b are potential targets for lung cancer treatment. Furthermore, QDs conjugated with Camellia sinensis leaf extract were shown to inhibit the lung tumor cell cycle and decrease cancer cell viability [51]. In another study, the treatment of uncapped CdTe nanoparticles (520Q and 580Q) induced oxidative stress in lung tumors and human bronchial epithelial cells. However, 730Q was shown to have a visible effect only after longer exposure times but not with shorter exposure times [52]. Aside from their many benefits, QDs comprise heavy metals such as cadmium, which is a known toxicant and carcinogen and poses conceivable hazards for clinical applications.

### 7.6. Graphene

Graphene is also referred to as isolated atomic layers of graphite and is a flat mesh of regular hexagonal rings. It is extremely thin, transparent, lightweight, and is a good conductor of heat and electricity, making it an attractive material for cancer imaging and detection. Graphene has many useful properties, including its bipolar transistor effect and large quantum oscillations [53], making it a good candidate for effective cancer imaging. Graphene also has a large specific surface area, which is useful for loading anticancer drugs due to the presence of π-π stacking and hydrophobic interactions [53]. Graphene oxide is an oxidized derivative of graphene and has been used in cancer treatment, drug delivery, and cellular imaging (Table 6). Graphene oxide is known to amplify the physicochemical properties of advanced functional materials. In one study, it was found that isotopic graphene-isolated Au nanocrystals with cellular Raman-silent signals could be used for cancer cell detection [54]. In another report, a graphene oxide-based switch-on fluorescent probe was used for glutathione detection and cancer diagnosis [55].

### 7.7. Fullerene

Fullerene belongs to the Buckminsterfullerene family and it is an allotrope of the carbon family, whose molecule contains carbon atoms interconnected by single or double bonds to form a mesh-like structure (Figure 5). Fullerene has been used in chemical applications, where drug molecules can be trapped in the fullerene mesh for successful drug delivery. Fullerene (C60) has been used in cancer diagnosis and detection. For example, fullerene has been successfully used in the development of biosensors to detect glucose levels in blood serum [59].

In another study, it was reported that helium (He) can also be trapped by heating C_60_ in helium vapor under pressure [60]. In a recent study, it was reported that fullerenes and metal nanomaterials can protect the human body from endogenously and exogenously produced damaging reactive oxygen species by using their reducing properties. It has been found that fullerenes and metal-based nanomaterials can selectively clear diseased cells in tissues and, consequently, avert the growth of chronic inflammatory diseases. Fullerenes and metal-based nanomaterials have tremendous potential in the treatment of aging-related disorders [60]. The application of different fullerene-based nanomaterials in cancer treatment is summarized in Table 7.

### 7.8. Carbon Nanotubes

Carbon nanotubes are made of single-wall carbon. They are excellent materials and have many useful applications due to their electrical and thermal conductivity, durability, and lightweight properties (Figure 6). Single-wall nanotubes have been found to exhibit superior properties compared to multi-wall nanotubes, as well as silver or copper materials. They have a diameter of 1 nanometer and the tube is very long. The physical properties of carbon nanotubes make them very important materials and they can be used in electronic nanodevices, composite materials, hydrogen storage, rechargeable lithium batteries, biosensors, and touch screens. In addition, carbon nanotube biosensors have been used to detect organophosphorous compounds such as in a study where gold nanoparticles and carbon nanotubes were deposited on a gold wire. The biosensor utilized acetylcholine-esterase enzymes on carboxylic single-walled carboxylic nanotubes, which were then pasted onto the heart electrode. The electrode was coated with Nafion to prevent enzyme leaching and become a sensor electrode. This nanosensor works on the principle of inhibiting the AChE enzyme [64,65]. In one study, carboxyl-functionalized carbon nanotubes were treated with a human T-cell cell line, causing the induction of caspase-2 gene expression in the cells. The results showed a slight decrease in the cell viability of cancer cells treated with carboxyl-functionalized carbon nanotubes. The results of the molecular analysis showed that Cas2 mRNA increased in the cancer cells treated with carboxyl-functionalized carbon nanotubes [66]. The application of carbon nanotube-based nanomaterials as carriers in cancer cells is summarized in Table 8. Carbon nanotubes have been used in cancer diagnosis and imaging due to their biocompatibility, thermodynamic properties, and varied functionalization [67,68].

### 7.9. Liposomes

Liposomes are nanoparticles composed of lipids. They consist of a closed spherical lipid bilayer forming an inner cavity that can hold aqueous solutions. The two lipid layers consist of two tightly arranged phospholipid sheets with a hydrophobic tail and a hydrophilic head area. The hydrophobic tails are attracted to each other, whereas the two membrane heads are directed to the surrounding water. This forms a double layer of phospholipid molecules, which prevents the inner solution from escaping to the outside. The solution can then be transferred with the liposome to where it is needed. They are novel in pharmaceutical drug delivery systems and drug delivery vehicles (Figure 7). Liposome nanoparticles have had a significant impact on chemotherapy as they can improve selectivity, decrease cytotoxicity, and enhance the solubility of hydrophobic drugs (Figure 7). Small liposomes with a diameter of approximately 50 nm can be prepared using microfluidics, whereas large liposomes with a diameter of 75 nm exhibited similar drug retention in in vitro and animal studies. Nevertheless, the degree of extravasation was dependent on the size of the liposomes, where small liposomes showed better tissue distribution than large liposomes [69]. The size of liposomes is critical. Several studies have investigated chemotherapy delivery using liposomes, including lipofectamine, which can increase transfection efficiency and reduce chemo-resistant proteins. For example, studies have utilized liposomes to deliver “Glucose regulated protein 78” or to target clustering of 1, 2-dioleoyloxy-3-trimethylammoniumpropane liposomes in cancer stem cells and breast cancer cells. The results showed the successful delivery of the composites in cancer stem cells and breast cancer cells [69]. In addition, magnetic liposomes have been used in cancer therapy [70], and liposomal IR-780, which is a highly stable nanotheranostic agent, has been used in the treatment of brain tumors [70].

### 7.10. Dendrimers

Dendrimers are highly defined artificial macromolecules with a three-dimensional network that have a high number of functional groups. Due to their size and structure consisting of dendritic arms or branches built around a linear polymer core, dendrimers have been used in nanomedicine. Dendrimers are useful as delivery or carrier systems for drugs and genes. They can also be used as antifungal, antibacterial, and anticancer agents (Figure 8). Dendrimers have three layers: the molecular core, the inner layer, and the outer layer. In addition, they have solubility, viscosity, and micellar properties. Due to their structure, they have multifunctional abilities, making them useful for building next-generation nanodevices for imaging and diagnosis [71]. There is a plethora of applications for dendrimers and they have played a significant role in the development of materials sciences. Some examples of the use of dendrimers in medical applications include drug delivery, drug encapsulation, drug conjugation, drug nanocarriers, anticancer agents, gene delivery agents, photodynamic therapy, medical theranostics, and biosensors [72]. Moreover, they have been used in anticancer drug delivery due to their higher stability, water solubility, and decreased immunogenicity. In addition, dendrimers can cause hypervascularization, increased permeability of cancer cells, and poor lymphatic drainage, which is beneficial for passive targeting. The beneficial effects of dendrimers can result in the selective accumulation of drug molecules in tumor tissues. Several studies have suggested that dendrimers should be included in the structure of drug molecules by encapsulation or conjugation, which could yield safer and more efficient medicines [53]. Dendrimers are synthesized using a process called divergent synthesis. This involves the stepwise addition of monomers to a central core, resulting in a highly branched structure. The monomers can be modified to change the properties of the dendrimer, such as its size, shape, and reactivity. The synthesis of dendrimers is a complex process and requires careful control of the reaction conditions. The properties of dendrimers can be tailored by varying the monomers used, the reaction conditions, and the number of monomers added. Dendrimers have a number of unique properties that make them useful for a wide range of applications. They are highly stable and resistant to degradation and they can be tailored to meet specific needs. They are also highly soluble and can be used to deliver drugs and other molecules to specific targets. The application of dendrimers in cancer detection and diagnosis is summarized in Table 9.

### 7.11. Nanostructure Lipid Carriers (NLCs)

The application of lipid-based nanocarriers has shown promising results in cancer diagnosis and treatment. Polymeric nanoparticles synthesized from a variety of lipid-based compounds or combined with vectors, such as liposomes, ethosomes, and transfersomes, may help drugs to overcome issues related to resistance to absorption in biological membranes. The combined effects of lipid-based nanocarriers are known to improve the efficacy and accuracy of polymeric nanoparticles [77,78]. In one study, the combined treatment of conatumumab-decorated, reactive oxygen species-sensitive irinotecan prodrug and quercetin co-loaded nanostructured lipid carriers was investigated for colorectal cancer treatment [20]. 

### 7.12. Other Nanoparticles and Their Application in Various Cancer Cells

The use of various types of nanoparticles has led to the expansion of nanotechnologies in numerous fields, such as pharmaceuticals, energy generation, chemicals, and drug delivery, as well as process efficiency improvements. There are several types of nanoparticles used today for detecting cancer cells and for nanotherapy. The size, kind, and structure of these nanoparticles can determine their functions [79]. After discussing the most common types of nanoparticles, it is clear that there are many others that are quite different from traditional particles such as amino acids and semiconductor nanoparticles, as seen in the table below. Additionally, there is potential for the application of nanorobots in cancer diagnosis and treatment. Nanorobots are controlled devices composed of nanometric components that can interact with the cellular membrane due to their small size, offering a direct channel to the cellular level. Nanorobots can improve treatment efficiency by performing advanced biomedical therapies using minimally invasive operations. Moreover, nanorobots are currently designed to recognize 12 different types of cancer cells [80]. The impact of nanoparticles on various types of cancer cells, such as colon cancer [26,27,30,31], glioblastoma and melanoma [26], neuroblastoma [28], breast cancer [29], cervical cancer [29], lung cancer [34,37], prostate cancer [35], oral squamous cell cancer [36], liver cancer [22], brain cancer [78], pancreatic cancer [81], bladder cancer [82], colorectal cancer [83], and cancer stem cells [23], has been investigated. The application of other types of nanoparticles in cancer therapy is summarized in Table 10.

### 7.13. Formulation of Nanomaterials for Drug Delivery

Conventional dosage methods, such as oral ingestion and intravenous (iv) injection, have some limitations, but controlled release systems offer many advantages. The use of controlled release systems not only improves drug efficacy and reduces side effects but also improves patients’ compliance with their medication. The implementation of these drug delivery systems has proven to be convenient for both doctors and patients, leading to broad market prospects [87]. Numerous controlled release systems, including nanoparticles [88], micelles [89], hydrogels [90], and electrospun membranes [91], have been investigated in recent years with the aim of satisfying a number of requirements for clinical applications. Poly (d,l-lactide-*co*-glycolide) (PLGA)-based mono/bicomponent electrospun polymer membranes have been approved by the US Food and Drug Administration (FDA) and have been used as an implant material for many years. The mechanical properties and degradation rate of these membranes can be tailored [91] by changing the ratio of lactic acid (LA) and glycolic acid (GA) in the copolymer. To adjust the swelling properties and control the release profile of the fibrous membrane, five kinds of polymers, all with FDA approval for implant application, were blended into the fibrous membrane as a second component. These polymers included poly(ethylene glycol) (PEG) and poly(ethylene glycol)-*b*-poly(d,l-lactide) (PEG-*b*-PDLLA) [91]. In general, hydrophilic polymer carriers allow a rapid release profile, whereas hydrophobic polymers release their loaded drugs very slowly. Multicomponent electrospun fibers consisting of both hydrophilic and hydrophobic polymers have been extensively investigated for drug delivery [92]. To maintain therapeutic concentrations of anticancer molecules for a relatively long time through a combination of burst and sustained drug release mechanisms, a hybrid of poly-caprolactone and gelatin (PCL/GEL) was used for the co-encapsulation of free curcumin (CUR) and CUR-loaded mesoporous silica nanoparticles [92].

### 7.14. Clinical Application of Nanomaterials in Cancer Treatment

Different types of nanomaterials have shown promising results in clinical situations. In order to safely and effectively deliver anticancer drugs to the targeted site in the cancer tissues, anticancer drugs are typically conjugated with nanomaterials. For example, CRLX101 is a novel nanoparticle–drug conjugate of camptothecin that was used in phase Ib/II clinical trials for patients with advanced rectal cancer. The results showed that CRLX101 was well tolerated with only one grade 4 toxicity (lymphopenia), and a pathologic complete response was seen in 19% of patients in the overall cohort and 33% of patients at the maximum tolerated dose [93]. In another clinical trial, the efficacy of the nanoparticle–drug conjugate CRLX101 in combination with bevacizumab was examined in metastatic renal cell carcinoma patients in a phase I-IIa clinical trial [94]. In addition, a first-in-human phase 1/2a trial of CRLX101, a cyclodextrin-containing polymer–camptothecin nanopharmaceutical, was conducted for patients with advanced solid tumor malignancies [95].

### 7.15. Current Challenges and Future Prospects

There are many advantages of magnetic nanoparticles, such as their nanosize, high reactivity, and better surface area, which can have lethal effects on human cells or tissues. It has been reported that nanoparticles can enter the human body during consumption or breathing and can transfer to various organs and tissues and exert their toxicological effects. The effect of nanoparticles on the environment is well known and there have been efforts to design safe nanomaterials. The use of nanoparticles, including dendrimers, may have catalytic properties that interfere with enzymatic tests or excessive charges that could impact the cell membrane or lead to cell death. The use of nanomaterials for cancer diagnosis and imaging has many advantages; however, there are also concerns regarding their potential toxicity and harmful effects on the human body [96]. Nanoparticles can accumulate in organs such as the liver and kidneys, leading to potential problems for the patient, and the extent to which these deposits affect the human body is not fully understood [97]. In the past, the major challenge for nanoparticles was crossing the different layers of the skin or passing through tiny blood vessels. However, today, scientists are concerned about nanoparticle toxicity and the risks to the environment and health, especially from metallic nanoparticles, which may interact with other contaminants and influence their toxicity.

The hazardous effects of nanomedicine are unlike those of classic medicines. The toxicology of particulate matter differs from that of chemical substances (which may or may not be dissolvable in matrices of biological systems), which significantly affects the possible exposure of different organs. Nanosized particles, on the other hand, can cross many barriers in the body, with the blood–brain barrier being an important one that may affect the brain in different ways. Similarly, the size of nanoparticles easily allows them to pass through cell membranes and intracellular organelles, as well as the nucleus [98].

The use of nanomedicine helps to balance the toxicity and efficiency of therapies. It integrates chemical, biological, and physical properties to explore their behavior in vivo. One of the critical factors in the translational evaluation of nanomedicine is assessing the biodistribution value of nanocompounds after administration in pre-clinical and clinical studies. Each technology has its own features, limitations, and capabilities for assessing real-time accumulation in cells, tissues, and organs [99,100].

To overcome the toxicity of some NPs, green nanotechnology has been developed as a desirable alternative due to its avoidance of harsh processes and factors such as high pressure, elevated temperatures, and harmful chemicals, as well as foreign stabilizing or coating agents. In this approach, many microorganisms and plants are incorporated in nanoparticle synthesis to obtain highly optimized, low-toxicity, eco–friendly nanoparticles that are safe for humans and the environment [101]. Additionally, the irregular dispersion properties of smart Ag nanoparticles make them resistant to bacteria and facilitate their accumulation, as their surface ligands resemble the membrane of bacteria, resulting in a remarkable antibacterial effect. This invention of boosted dispersion properties provides excellent application opportunities for smart nanotechnologies in various fields [102].

Nanoparticles have the ability to cross many barriers in the body, including the blood–brain barrier, which can affect the brain in different ways. Similarly, their nanosize allows them to easily pass through cell membranes and intracellular organelles, as well as the nucleus. Nanomedicine drug delivery can utilize a wide range of biological substances or chemical structures, as previously discussed It is important to note that the potential for interactions with cells that can lead to toxicity depends heavily on the actual structure of the nanoparticle formula [103].

As mentioned previously, one of the critical factors in the translational evaluation of nanomedicine is assessing the biodistribution value of the nanocompounds after administration in pre-clinical and clinical studies. Studies have shown that each technology has its own features, limitations, and capabilities for assessing real-time accumulation in cells, tissues, and organs [104]. Some studies have shown that nanoparticles have a tendency to be deposited on the skin, which has been demonstrated in both human and porcine in vitro and in vivo studies [105]. Heavy mineral nanoparticles, such as tin, mercury, and lead, are known to be stable and have low degradation, which can cause high environmental toxicity [106]. Studies have assessed the systemic toxicity, genotoxicity, and carcinogenic effects in rats exposed to both tannic acid and iron (III) nanoparticles [107], as well as in primary and cancerous human cells [108]. The future of nano-based materials for cancer diagnosis is to develop safe nanomaterials with minimal toxicity and better biodistribution.

## Figures and Tables

**Figure 1 pharmaceutics-15-01025-f001:**
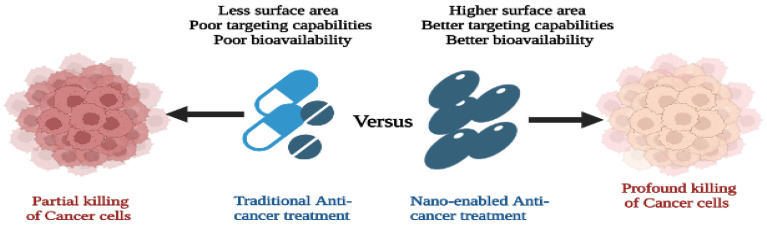
Comparison of traditional and nano-based methods of cancer treatment.

**Figure 2 pharmaceutics-15-01025-f002:**
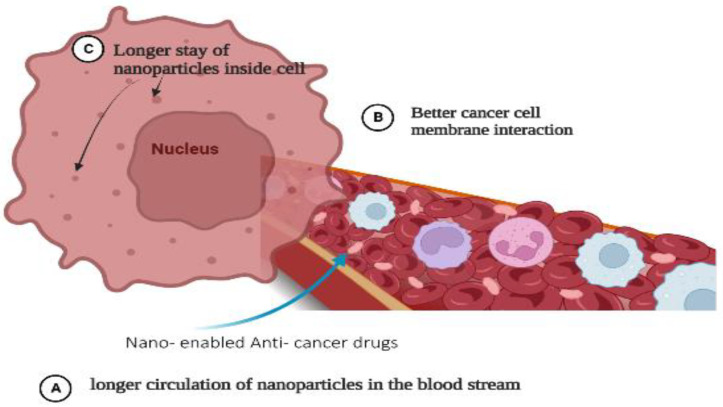
Entry and movement of nano-enabled anticancer drugs in a cancer cell.

**Figure 3 pharmaceutics-15-01025-f003:**
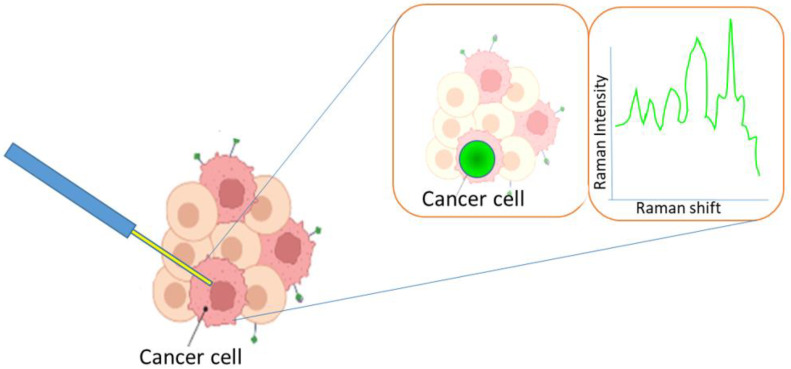
Application of Raman technique in cancer diagnosis.

**Figure 4 pharmaceutics-15-01025-f004:**
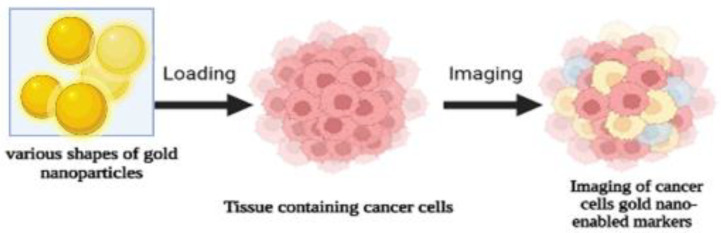
Applications of gold nanoparticles in cancer imaging and detection.

**Figure 5 pharmaceutics-15-01025-f005:**
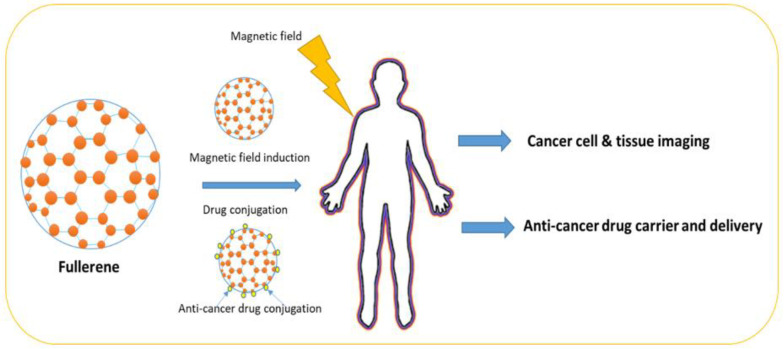
Fullerene structure and shape.

**Figure 6 pharmaceutics-15-01025-f006:**
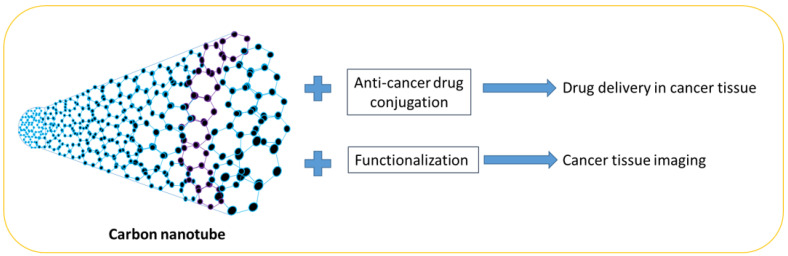
Carbon nanotubes and their application in cancer drug delivery and imaging.

**Figure 7 pharmaceutics-15-01025-f007:**
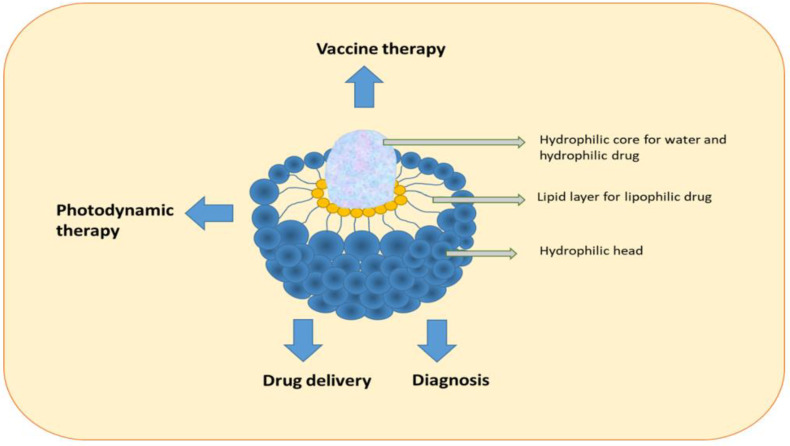
Structure and applications of liposomes.

**Figure 8 pharmaceutics-15-01025-f008:**
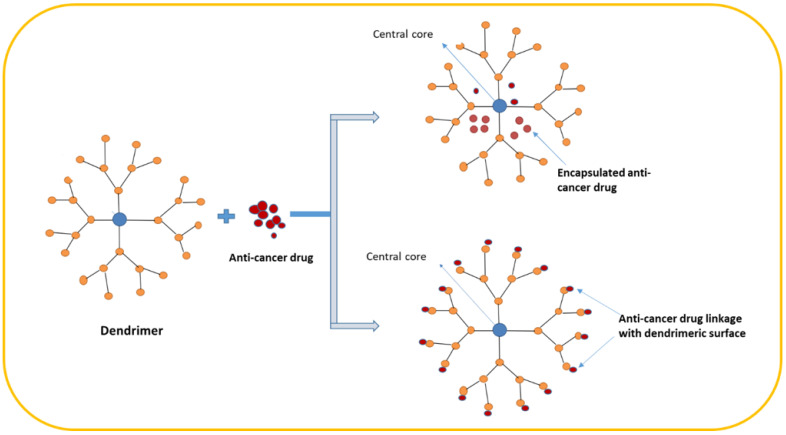
The structure and application of dendrimers in drug encapsulation and conjugation.

**Table 1 pharmaceutics-15-01025-t001:** Application of nanoparticles in cancer diagnosis [16].

Type of Nanoparticles	Application of Nanoparticles	Properties	Example of Cancer Diagnosis
1. Carbon-based nanoparticles	Used for cancer detection and diagnosis	Excellent physio-chemical properties, including high-level penetration into the cell membrane, high surface area, and high capacity for drug loading	Both in vivo and in vitro studies show that nanodroplets are an effective contrast material for both photoacoustic and ultrasound imaging
2. Ceramic nanoparticles	For better drug delivery and cancer imaging	High biocompatibility	Clinical studies on gold NS-based photothermal therapy are under consideration for ablating repetitive head and neck tumors, as well as cancer imaging
3. Metallic nanoparticles	Detection and imaging of cancer cells/tissues	Magnetic nanoparticles are crucial for metastatic breast cancer detection and protection	Gold nanoparticles for Raman imaging
4. Polymeric nanoparticles	Drug delivery and diagnostics	Surrounded by a polymer shell	Block copolymer-coated nanoparticles (TPIONPs) connected with RGD peptides and dye molecules to target tumors
5. Lipid-based nanoparticles	Use as a drug carrier and drug delivery system for cancer diagnosis	Better biocompatibility and low toxicity in comparison with inorganic nanoparticles	Conjugation of anti-HER2 antibodies on phospholipid-coated QDs revealed the ability to target HER2-positive tumors

**Table 2 pharmaceutics-15-01025-t002:** Application of platinum-based nanoparticles in various types of cancer treatment.

Cancer	Nanoparticle Type	Application	Results	Citation
Glioblastoma and melanoma cells	Ag-Pt nanoparticles	Treating cancer cell glioblastoma and melanoma	No cytotoxic effects in healthy cells (in vitro)	[26]
Colon cancer cell lines	Platinum nanoparticles (Pt NPs)	Microscopy images of colon cancer cells for better imaging	Cancer cell mortality increased to 62% (in vitro)	[27]
Neuroblastoma cancer	PtNPs and RA nanoparticles	Induction of cancer cell death	PtNPs and RA nanoparticles induced cancer cell death due to apoptosis, as well as oxidative DNA damage (in vitro)	[28]
Breast and HeLa cervical cancer lines	Biogenic-platinum nanoparticles	Application as an antimicrobial and anticancer agent	Development of a potential antibacterial and anticancer agent (in vitro)	[29]
Colon cancer cells	Gold-decorated platinum and palladium nanoparticles	Improved the effectiveness of simulated anticancer proton therapy	Induced cancer cell death due to apoptosis (in vitro)	[30]
Colon cancer cells	Pt/MgO nanoparticles	Induced cancer cell death in colon cancer cells	Downregulation of Bcl2 in colon cancer cells. Upregulation of Bax and p53 in colon cancer cells (in vitro)	[31]

**Table 3 pharmaceutics-15-01025-t003:** Application of gold nanoparticles in cancer imaging and diagnosis.

Nanoparticles	Nanoparticle Type	Imaging Application	Advantage	Citation
Gold nanoparticles	Spheres	X-ray imaging	High payload delivery	[32]
	rods, shells, labeled spheres	Fluorescence imaging		
	Spheres, stars	Surface-enhanced Raman spectroscopy imaging	Generates a strong electromagnetic field	
	Spheres, clusters, rods, shells	Photoacoustic imaging	Requires strong absorption in the NIR window	
	Primarily spheres	Optical imaging	The light used is in the NIR window	

**Table 4 pharmaceutics-15-01025-t004:** Impact of gold nanoparticles on different cell types.

Type of Cell	Application	Results	Citation
microRNA biology	Detection of miRNA-155	It can detect cancer under optimum experimental conditions (in vitro, in vivo)	[13]
Tumor	Drug delivery for cancer imaging	Inert nanoparticle surfaces enable better imaging by reducing protein absorption (in vitro)	[33]
Lung cancer biomarker	GNP crosslinked with hnRNPB1, thiol as crosslinker	hnRNPB1 biomarker for cancer diagnosis (in vitro)	[34]
Prostate-specific antigen (PSA)	Immunosensor-based nanomaterial	Immunosensor with high sensitivity, selectivity, and long-term stability for cancer bioassay analysis (in vitro)	[35]
Oral squamous cell carcinoma	Nano-ELISA associated with gold nanorod assay	ELISA improved the sensitivity of cancer analyses (in vitro, in vivo)	[36]
Cell lung cancer (NSCLC)	Development of immuno-sensor	Better diagnosis of CYFRA21-1 cancer detection (in vitro, in vivo)	[37]
Detection of cancer biomarker CEA	Fluorescence of FITC via FRET	Fluorescence FITC using the FRET technique can detect cancer cells with better selectivity (in vitro, in vivo)	[21]

**Table 6 pharmaceutics-15-01025-t006:** Application of graphene-based nanomaterials in cancer treatment.

Type of Nanomaterial	Use	Results	Citation
Graphene quantum dots (doxorubicin) (DOX)	Drugs against blood cancer cells	In vitro results showed no significant toxicity against blood cancer cells (in vitro)	[56]
Graphene oxide (GO) (doxorubicin, DOX) and hydrophobic (Methotrexate MTX)	MTT assay for cytotoxicity of GCANBN	Nanomaterials can successfully deliver drugs as nanocarriers (in vitro)	[57]
Graphene quantum dots (GQDs)	Graphene quantum dot (GQDs)-Fe_3_O_4_@C@TDGQDs microspheres for drug delivery	In vitro results showed that Fe_3_O_4_@C@TDGQDs microspheres are safer materials for drug delivery of cancer drugs (in vitro)	[58]

**Table 7 pharmaceutics-15-01025-t007:** Application of fullerene-based nanomaterials in cancer treatment.

Fullerene Type	Application	Function	Citation
Fullerene (C60)	Biosensor fabrication	Successful development of biosensors to detect glucose levels in blood serum (in vitro, in vivo)	[61]
Fullerene (C60)	In vitro cytotoxic activity of C60 + LA nanocomplex by MTT assay	C_60_ + LA nanocomplex showed higher cytotoxicity toward cancer cells (in vitro)	[62]
C3	In vitro testing of C3 for clonogenic cancer detection	C3 protected GM-CFC in a concentration-dependent manner (in vitro)	[63]

**Table 8 pharmaceutics-15-01025-t008:** Application of carbon nanotube-based nanomaterials as carriers in cancer cells.

Nanoparticle	Application	Results	Citation
Carbon nanotubes and boron nitride nanotubes (BNNT)	Drug carbon nanotube and boron nitride nanotube (BNNT)-based carriers	Carbon nanotubes are superior to CNT as nanocarriers of the 6-TG drug (in vitro)	[67]
SWCNT + B3LYP and M06-2X	Drug delivery and as a drug carrier	The functionalization of SWCNT has increased the drug solubility in an aqueous solution (in vitro)	[68]

**Table 9 pharmaceutics-15-01025-t009:** Application of dendrimers in cancer detection and cancer diagnosis.

Complex	Function	Application	Results	Citation
Dendrimer generation CLs and (PG4)	Breast cancer treatment	Cell cycle analysis	Dendrimers can be used for carrier delivery in breast cancer therapy (in vitro)	[73]
Cisplatin (CDDP) and human antigen R (HuR)-	Encapsulating chemotherapeutic drugs	Drug delivery	Developing multifunctional dendrimer-based nanoparticles (in vitro)	[74]
5-aminolevulinic acid (ALA) dendrimers	Bladder cancer	Diagnosis	Fluorescence diagnosis of bladder cancer: a novel in vivo approach (in vivo)	[75]
PAMAM G4.5 dendrimers	Breast cancer	Diagnosis	In vitro and in vivo uptake studies	[76]

**Table 10 pharmaceutics-15-01025-t010:** Application of other types of nanoparticles in cancer therapy.

Nanoparticle	Cancer Type	Application	Citation
miRNA	Colorectal cancer (CRC)	It can be used as a circulating biomarker for the early diagnosis of CRC (in vitro)	[84]
Raman-active nanoprobe (RAN	Circulating cancer stem cells (CCSCs)	Better imaging using the Raman imaging method to detect cancer cells (in vitro, in vivo)	[85]
Fumed silica nanoparticles	Detecting cancer pathways	Nanoparticles can successfully bind to multi-site phosphorylated peptides for better cancer detection (in vitro, in vivo)	[86]

## Data Availability

Data will be available upon request.

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
