# Peer review of "Nanotechnology in Cancer Diagnosis and Treatment"

_pharmaceutics, 2023, doi:10.3390/pharmaceutics15031025_

Round 1

Reviewer 1 Report

Noor Alrushaid et al. gave a comprehensive review on recent advances in diverse nanoparticles for cancer diagnosis and treatment. Actually, it is very hot topic and remains great challenges. However, at present, this review was not organized well and some contents were in chaos. In particular, there were no comparisons among nanoparticles as-mentioned when they were fabricated for cancer diagnosis and treatment. In addition, the following points should be considered. After improvement, I think it could meet the standard of publication for this journal.

1. Many many spelling errors appear in the whole manuscript. The authors should correct them very carefully.

2. In Table 1, many properties of nanomaterials were not associated with cancer diagnosis and treatment.

3.    If possible, the authors give a graphical abstract to demonstrate the main content of this review.

4.     In the part of Introduction, some updating reports on metal nanoclusters for cancer diagnosis and treatment are recommended to be cited, such as Anal. Chem. 2022, 94, 3023-3026; Sci. Adv. 2020, 6, eabb1421.

5.     To improve the quality of this review, the authors add more future perspective and challenges in the last part.

Author Response

  1. Many many spelling errors appear in the whole manuscript. The authors should correct them very carefully.

Response: Thank you for your comments. Spelling errors have been corrected in the entire manuscript.

  1. In Table 1, many properties of nanomaterials were not associated with cancer diagnosis and treatment.

Response: Thank you for your comments. Table 1 has been revised by mentioning the properties of the nanomaterials cancer diagnosis

  1.   If possible, the authors give a graphical abstract to demonstrate the main content of this review.

Response: Thank you for your comments. Graphical abstract is added in the revised manuscript.

  1. In the part of Introduction, some updating reports on metal nanoclusters for cancer diagnosis and treatment are recommended to be cited, such as Anal. Chem. 2022, 94, 3023-3026; Sci. Adv. 2020, 6, eabb1421.

Response: Thank you for your comments. Following references are added in the introduction as suggested.

  • Wang X, Li J, Zhang W, Li P, Zhang W, Wang H, Tang B. Evaluating diabetic ketoacidosis via a MOF sensor for fluorescence imaging of phosphate and pH. Chem Commun (Camb). 2022 Mar 1;58(18):3023-3026. doi: 10.1039/d1cc06876h. PMID: 35156674.
  • Zhang W, Wang X, Li P, Zhang W, Wang H, Tang B. Evaluating Hyperthyroidism-Induced Liver Injury Based on In Situ Fluorescence Imaging of Glutathione and Phosphate via Nano-MOFs Sensor. Anal Chem. 2020 Jul 7;92(13):8952-8958. doi: 10.1021/acs.analchem.0c00925. Epub 2020 Jun 8. PMID: 32438804.
  • Gao L, Zhang Y, Zhao L, Niu W, Tang Y, Gao F, Cai P, Yuan Q, Wang X, Jiang H, Gao X. An artificial metalloenzyme for catalytic cancer-specific DNA cleavage and operando imaging. Sci Adv. 2020 Jul 15;6(29):eabb1421. doi: 10.1126/sciadv.abb1421. PMID: 32832637; PMCID: PMC7439319.
  1. To improve the quality of this review, the authors add more future perspective and challenges in the last part.

Response: Thank you for your comments. We have revised the current challenges and future prospective in the revised manuscript.  

Reviewer 2 Report

The authors of this study tried to compile a review of nanotechnology, cancer diagnosis, and treatments. Although the title of the manuscript was intriguing, as expected, it needed to be presented constructively. Cancer diagnosis and treatment, like nanotechnology, is a vast field in and of itself. The title seems created from the perspective of broad subject areas, each of which has multiple streamlines, thus requiring numerous latest articles survey with more references; however, it could have been narrowed and focused on any of one interest.

Nonetheless, throughout the manuscript, several sections need to be improved. For example, in the introduction, it is hard to understand what the authors are trying to address between all three paragraphs and is inconclusive; similar observations are noticed in the other few following sections. Moreover, some repeated paragraphs are seen in section 5.11. In addition, the figures are not giving any information, and most of them need to be labeled clearly; thus, it lacks understanding for the readers. Finally, the tables could have been improved in providing necessary and relevant information, such as the nanoparticle concentration, the study type (in-vitro or in-vivo), and so on.

Overall, the current version of this manuscript lacks in addressing the current problems and the research gap between each method and technique utilized in nanotechnology toward cancer diagnosis and treatments, and the relevant suggestions to the issues. Hence, the current version of this manuscript is unsuitable for publication, and it is recommended to submit an updated or modified version to the relevant journals of MDPI in Biomedicine or Nanomaterials.

Author Response

The authors of this study tried to compile a review of nanotechnology, cancer diagnosis, and treatments. Although the title of the manuscript was intriguing, as expected, it needed to be presented constructively. Cancer diagnosis and treatment, like nanotechnology, is a vast field in and of itself. The title seems created from the perspective of broad subject areas, each of which has multiple streamlines, thus requiring numerous latest articles survey with more references; however, it could have been narrowed and focused on any of one interest.

Nonetheless, throughout the manuscript, several sections need to be improved. For example, in the introduction, it is hard to understand what the authors are trying to address between all three paragraphs and is inconclusive; similar observations are noticed in the other few following sections.

Response: Thank you for your comments. We have modified the introduction to be focused on the role of different types of nanomaterials in cancer diagnosis

Moreover, some repeated paragraphs are seen in section 5.11. In addition,

Response: Thank you for your comment. Section 5.11 was reorganized and corrected in the revised manuscript

the figures are not giving any information, and most of them need to be labeled clearly; thus, it lacks understanding for the readers.

Response: Thank you for your comments. We have total 8 figures, the idea is to provide concise information in the figures so that they are easy for the readers to understand the significance but thank you for your suggestion.

Finally, the tables could have been improved in providing necessary and relevant information, such as the nanoparticle concentration, the study type (in-vitro or in-vivo), and so on.

Response: Thank you for your comments. All the tables have been improved by adding the in vitro and in vivo information

Reviewer 3 Report

1)      Rewrite the introduction section as it fails to give the background of the manuscript. The introduction to nanoparticles for current utilization in clinical studies needs to mention.

2)      Highlight the advantages of nanoparticles over other anticancer treatment methods.

3)      Elaborate the nanoparticles synthesis section according to types.

4)      The mechanism of drug release from NPs needs to be explained.

5)      Diagnosis and therapeutic application of NPs are not elaborated. Explain this application with suitable examples.

6)      Only nanoparticles are referred to as a nanotechnological platform and other nanoformulation and their utilizations in missing. For instance, the role of nanorobots in cancer treatment needs to be highlighted.

https://doi.org/10.1016/j.jddst.2023.104173

Moreover, nanostructured lipid carriers (NLCs) are being widely used in cancer treatment.

https://doi.org/10.1016/j.jddst.2021.102957. Add all this information for a more good review article to cover all revel vent sub-topic.

 8)      NPs synthesis methods need to explain alongside suitable examples.

9)      NPs utility in different types of cancer is not explained.

10)     Future Prospectives need to be added at the end

12)     Conclusive remarks of the manuscript need to be summed up with the current understanding and future outlook.

13) More focus should be provided on the clinical application of NPs for cancer treatment with all the latest applications for more authenticity of the data.

Author Response

1)      Rewrite the introduction section as it fails to give the background of the manuscript. The introduction to nanoparticles for current utilization in clinical studies needs to mention.

Response: Thank you for your comments. The introduction has been reorganized and focused on the application of nanomaterials in cancer diagnosis

2)      Highlight the advantages of nanoparticles over other anticancer treatment methods.

Response: Thank you for your comment. The advantages of nanomaterials has been highlighted in separate paragraph

3)      Elaborate the nanoparticles synthesis section according to types.

Response: Thank you for your comment. We have briefly described nanomaterials

4)      The mechanism of drug release from NPs needs to be explained.

Response: Thank you for your comments. We have describe the mechanism of drug from NPs in the separate section 7.11

5)      Diagnosis and therapeutic application of NPs are not elaborated. Explain this application with suitable examples.

Response: Thank you for your comments. We have discussed the significance and application of NPs in the topic 3. Significance of Nanoparticles in Cancer Diagnosis 

6)      Only nanoparticles are referred to as a nanotechnological platform and other nanoformulation and their utilizations in missing. For instance, the role of nanorobots in cancer treatment needs to be highlighted.

https://doi.org/10.1016/j.jddst.2023.104173

Moreover, nanostructured lipid carriers (NLCs) are being widely used in cancer treatment.

https://doi.org/10.1016/j.jddst.2021.102957. Add all this information for a more good review article to cover all revel vent sub-topic.

Response: Thank you for your comments. We have added a separate paragraph on nanostructured lipid carriers (NLSs) and also added suggested references and other references in the revised manuscript.

 8)      NPs synthesis methods need to explain alongside suitable examples.

Response: Thank you for your comments. This review is mainly focusing on the application of NPs in cancer diagnosis/treatment, however, we have briefly described the methods of NPs synthesis in section 4. Synthesis of nanomaterials

9)      NPs utility in different types of cancer is not explained.

Response: Thank you for your comments. We have described the application of different types of cancer cells in section 7.11

10)     Future Prospectives need to be added at the end

Response: Thank you for your comments. We have provided future prospective in the revised manuscript

12)     Conclusive remarks of the manuscript need to be summed up with the current understanding and future outlook.

Response: Thank you for your comments. We have revised the current challenges in the revised manuscript. 

13) More focus should be provided on the clinical application of NPs for cancer treatment with all the latest applications for more authenticity of the data.

Response: Thank you for your comments. We have added separately discussed the application of NPs in the clinical investigations as described in 7.12 section

Reviewer 4 Report

Overview and general recommendation:

In this manuscript, the authors show that nanoparticles (NPs) are benefiting in many cancer-related areas such as cancer imaging, cancer diagnosis, drug delivery and cancer treatment because of higher surface area, better selectivity, better bioavailability. The authors talk about commonly used nanoparticles in the cancer diagnosis. The advantages and limitations are listed and the ways to overcome the limitations are also summarized here.

Overall the manuscript is OK. The inner logic of the review is clear and all the parts are organized properly. The tables are well organized and presented in an appropriate way. There are many parts which can improved. There are too many grammar mistakes in the manuscript, please find someone who is expertise in English to review. Some more references should be included in some parts. The combination application of different NPs should be talked in the manuscript. And I suggest the authors strengthen the importance of this study in the manuscript.

Major comments:

1.    This manuscript needs to be VERY VERY VERY carefully reviewed by someone who is expertise on scientific English, including the part that I already list some minor mistakes. Minor grammar mistakes are not listed for “5. Commonly used nanoparticles in the Cancer Diagnosis” since there are so many minor mistakes that it will take me a lot of time in fixing this. You should pay particular attention to grammar and sentence structure.

2.    In “Polymeric-based nanoparticles”, “Magnetic nanoparticles” and “Dendrimers”, too few examples or references are included, I suggest the authors include more references in these parts.

3.    The authors talk about anti-microbial effect of graphene. How about other types of NPs?

4.    It seems that Graphene, Fullerene and Carbon nanotube are all derived from carbon, but different atoms’ arrangements or configurations result in different application. Can the authors explain how different configurations generate different functions?

5.    In some cases, the combination of different NPs will work better than a single NP.  I suggest the authors add a part in section 5 talking about how the combination of different NPs will benefit in cancer imaging, diagnosis and treatment.

Minor comments:

1.    Page1 line13, I think it should be “…being diagnosed traditionally by application of nanoparticles…”

2.    Page1 line15, I think it should be “…higher surface area, higher volume proportion, better targeting...”

3.    Page1 line40, I think it should be “…in the body and movement from organs…”

4.    Page1 line42, I think ti should be “…to deliver drug…”

5.    Page2 ling44, I think it should be ”…like treatment and diagnose of cancers…”

6.    Page2 ling46, I think it should be ”…development of novel approaches…”

7.    Page2 ling47, I think it should be ” …to target specific molecules in the cancer…”

8.    Page2 ling51, I think it should be ” …helping to improve the cancer diagnosis…”

9.    Page2 ling55, I think it should be ” …the bottom-up method are mostly applied…”

10. Page2 ling70, I think it should be ” … is remaining to be a challenge in clinical diagnosis…”

11. Page2 ling76, I think it should be ” …livers and kidneys…”

12. Page2 ling81, I think it should be ” …and also discussed the challenges and shortcomings currently we face regarding….”

13. Page2 ling85, I think it should be ”… where the tumor locates…”

14. Page2 ling86, I think it should be ” …illness in cells…”

15. Page2 ling93, I think it should be ” …and thus helping to…”

16. Page2 ling94, I think it should be ” …many doctors with this technology….”

17. Page2 ling95, I think it should be ” … or proteins, fight the tumor …”

18. Page3 ling97, I think it should be ” … detected, which can guide…”

19. Page3 ling99, I think it should be ” … can be early detected by using…”

20. Page3 ling103, I think it should be ” … has been developed for the…”

21. Page3 ling111, I think it should be ” … difficult to be examined by other …”

22. Page3 ling112, I think it should be ” … traditional methods of…”

23. Page4 ling139, I think it should be ” … particle’s function in diagnosis…”

24. Page4 ling140, I think it should be ” … and are considered to be…”

25. Page4 ling142, I think it should be ” … and it use the magnetic nanoparticles…”

26. Page4 ling142-144, please fix this sentence, I can hardly understand it.

27. Page4 ling145, I think it should be ” … is IONPs, which is used to track…”

28. Page4 ling146, I think it should be ” … nanoparticles are used…”

29. Page4 ling151, I think it should be ” … patients. Colonoscopy is the most widely used…”

30. Page4 ling154, I think it should be ” … but it is found that…”

31. Page4 ling155, I think it should be ” … other techniques such as…”

32. Page4 ling156, I think it should be ” … but this application shows…”

33. Page4 ling159, I think it should be ” … or the urine by checking the level…”

34. Page4 ling162, I think it should be ” … different imaging applications such as…”

35. Page4 ling165, I think it should be ” … applications to detect…”

36. Page4 ling165, I think it should be ” … This imaging method allowing better guidance…”

37. Page5 ling172, I think it should be ” … specific carriers…”

38. Page5 ling182, I think it should be ” … These are EVs which are less toxic…”

39. Page5 ling186, I think it should be ” … The application of…”

40. Page13, the font of “5.9. Liposomes” should be fixed.

Author Response

In this manuscript, the authors show that nanoparticles (NPs) are benefiting in many cancer-related areas such as cancer imaging, cancer diagnosis, drug delivery and cancer treatment because of higher surface area, better selectivity, better bioavailability. The authors talk about commonly used nanoparticles in the cancer diagnosis. The advantages and limitations are listed and the ways to overcome the limitations are also summarized here.

Overall the manuscript is OK. The inner logic of the review is clear and all the parts are organized properly. The tables are well organized and presented in an appropriate way. There are many parts which can improved. There are too many grammar mistakes in the manuscript, please find someone who is expertise in English to review. Some more references should be included in some parts. The combination application of different NPs should be talked in the manuscript. And I suggest the authors strengthen the importance of this study in the manuscript.

Response: Thank you for your comments. The grammatical and spell errors have been corrected in the revised manuscript.

Major comments:

  1. This manuscript needs to be VERY VERY VERY carefully reviewed by someone who is expertise on scientific English, including the part that I already list some minor mistakes. Minor grammar mistakes are not listed for “5. Commonly used nanoparticles in the Cancer Diagnosis” since there are so many minor mistakes that it will take me a lot of time in fixing this. You should pay particular attention to grammar and sentence structure.

Response: Thank you for your comments. The manuscript has been corrected grammatically.

  1. In “Polymeric-based nanoparticles”, “Magnetic nanoparticles” and “Dendrimers”, too few examples or references are included, I suggest the authors include more references in these parts.

Response: Thank you for your comments. We have included more references in the revised manuscript.

  1. The authors talk about anti-microbial effect of graphene. How about other types of NPs?

Response: Thank you for your comments. We have rewritten the paragraph by focusing on the cancer diagnosis, not the anti-microbial action.

  1. It seems that Graphene, Fullerene and Carbon nanotube are all derived from carbon, but different atoms’ arrangements or configurations result in different application. Can the authors explain how different configurations generate different functions?

Response: Thank you for your comments. We have described graphene, fullerene and carbon nanotube considering the structural difference as the shape of nanoparticles is critical in cancer diagnosis.

  1. In some cases, the combination of different NPs will work better than a single NP.  I suggest the authors add a part in section 5 talking about how the combination of different NPs will benefit in cancer imaging, diagnosis and treatment.

Response: Thank you for your comments. We have added one paragraph discussing the importance of nanocompositesm (multiple nanoparticles) in cancer diagnosis.

Minor comments:

  1. Page1 line13, I think it should be “…being diagnosed traditionally by application of nanoparticles…”

Thank you for your comments.

Response: Corrected in the revised manuscript.

  1. Page1 line15, I think it should be “…higher surface area, higher volume proportion, better targeting...”

Response: Corrected in the revised manuscript.

  1. Page1 line40, I think it should be “…in the body and movement from organs…”

Response: Corrected in the revised manuscript.

  1. Page1 line42, I think ti should be “…to deliver drug…”

Response: Corrected in the revised manuscript.

  1. Page2 ling44, I think it should be ”…like treatment and diagnose of cancers…”

Response: Corrected in the revised manuscript.

  1. Page2 ling46, I think it should be ”…development of novel approaches…”

Response: Corrected in the revised manuscript.

  1. Page2 ling47, I think it should be ” …to target specific molecules in the cancer…”

Response: Corrected in the revised manuscript.

  1. Page2 ling51, I think it should be ” …helping to improve the cancer diagnosis…”

Response: Corrected in the revised manuscript.

  1. Page2 ling55, I think it should be ” …the bottom-up method are mostly applied…”

Response: Corrected in the revised manuscript.

  1. Page2 ling70, I think it should be ” … is remaining to be a challenge in clinical diagnosis…”

Response: Corrected in the revised manuscript.

  1. Page2 ling76, I think it should be ” …livers and kidneys…”

Response: Corrected in the revised manuscript.

  1. Page2 ling81, I think it should be ” …and also discussed the challenges and shortcomings currently we face regarding….”

Response: Corrected in the revised manuscript.

  1. Page2 ling85, I think it should be ”… where the tumor locates…”

Response: Corrected in the revised manuscript.

  1. Page2 ling86, I think it should be ” …illness in cells…”

Response: Corrected in the revised manuscript.

  1. Page2 ling93, I think it should be ” …and thus helping to…”

Response: Corrected in the revised manuscript.

  1. Page2 ling94, I think it should be ” …many doctors with this technology….”

Response: Corrected in the revised manuscript.

  1. Page2 ling95, I think it should be ” … or proteins, fight the tumor …”

Response: Corrected in the revised manuscript.

  1. Page3 ling97, I think it should be ” … detected, which can guide…”

Response: Corrected in the revised manuscript.

  1. Page3 ling99, I think it should be ” … can be early detected by using…”

Response: Corrected in the revised manuscript.

  1. Page3 ling103, I think it should be ” … has been developed for the…”

Response: Corrected in the revised manuscript.

  1. Page3 ling111, I think it should be ” … difficult to be examined by other …”

Response: Corrected in the revised manuscript.

  1. Page3 ling112, I think it should be ” … traditional methods of…”

Response: Corrected in the revised manuscript.

  1. Page4 ling139, I think it should be ” … particle’s function in diagnosis…”

Response: Corrected in the revised manuscript.

  1. Page4 ling140, I think it should be ” … and are considered to be…”

Response: Corrected in the revised manuscript.

  1. Page4 ling142, I think it should be ” … and it use the magnetic nanoparticles…”

Response: Corrected in the revised manuscript.

  1. Page4 ling142-144, please fix this sentence, I can hardly understand it.

Response: Corrected in the revised manuscript.

  1. Page4 ling145, I think it should be ” … is IONPs, which is used to track…”

Response: Corrected in the revised manuscript.

  1. Page4 ling146, I think it should be ” … nanoparticles are used…”

Response: Corrected in the revised manuscript.

  1. Page4 ling151, I think it should be ” … patients. Colonoscopy is the most widely used…”

Response: Corrected in the revised manuscript.

  1. Page4 ling154, I think it should be ” … but it is found that…”

Response: Corrected in the revised manuscript.

  1. Page4 ling155, I think it should be ” … other techniques such as…”

Response: Corrected in the revised manuscript.

  1. Page4 ling156, I think it should be ” … but this application shows…”

Response: Corrected in the revised manuscript.

  1. Page4 ling159, I think it should be ” … or the urine by checking the level…”

Response: Corrected in the revised manuscript.

  1. Page4 ling162, I think it should be ” … different imaging applications such as…”

Response: Corrected in the revised manuscript.

  1. Page4 ling165, I think it should be ” … applications to detect…”

Response: Corrected in the revised manuscript.

  1. Page4 ling165, I think it should be ” … This imaging method allowing better guidance…”

Response: Corrected in the revised manuscript.

  1. Page5 ling172, I think it should be ” … specific carriers…”

Response: Corrected in the revised manuscript.

  1. Page5 ling182, I think it should be ” … These are EVs which are less toxic…”

Response: Corrected in the revised manuscript.

  1. Page5 ling186, I think it should be ” … The application of…”

Response: Corrected in the revised manuscript.

  1. Page13, the font of “5.9. Liposomes” should be fixed.

Response: Corrected in the revised manuscript.

Round 2

Reviewer 1 Report

The authors have given an excellent response to my concerning points. Now, I think it could be accepted completely.

Author Response

Reviewer # 1

The authors have given an excellent response to my concerning points. Now, I think it could be accepted completely.

Thank you so much

Reviewer 2 Report

The authors revised most of the sections and were satisfied with the content provided within the study's scope. However, the manuscript required further improvements. The consistency of nanoparticles (NPs) in the introduction and the following sections must be maintained, and one should avoid new nanoparticles which are not discussed in detail. In addition, some areas are highlighted outside this study’s scope.  The comments and suggestions are listed below.

The authors must address missing acronyms and typos throughout the manuscript.  

I listed a few typos along with others, 43, 70, 81, 97, 121, 127, 160, 168, 177, 189, 197, 227, 230, 234, 236, 237, 267, 335, 350, 352, 360, 365, 380, 406, 469, 625, 628, 648, 652, 659, 664, 676, 677, 678, etc. In addition, most of the tables have typos and should correct accordingly.  

1.     The sentence from lines 22-25 can be rewritten in the abstract according to the graphical abstract.  

2.     The sentence from lines 28-30, what are many nanomaterials? This sentence could be rewritten.

3.     The sentence from lines 38-42 is too big and could be split into two sentences.

4.     What is “this technology” in line 87? Again, this sentence could be rewritten.

5.     References or origin of the data source need to be included in lines 92-95.

6.     As mentioned, NPs could detect early-stage 0/1 cancers (line 112). Stating this type of contradictory statement leads to possible questions. What kind of NPs and targeting agents are used to detect cancer, and what type of cancers? Can these NPs cross the blood-brain barrier (BBB) to detect brain cancers?

7.     The NP’s mono-nuclear phagocytosis or the reticule-endothelial system escape mechanism could have been highlighted in the introduction or before discussing NPs’ increased blood circulation in the body (lines 114 & 130).

8.     In line 116, “Moreover, nanoparticles can bind with the tumor,” what type of NPs and how could these NPs bind to cancer or tumor cells? Are they using any targeting agents modified to NPs? These sentences could be corrected accordingly.

9.     How could the NPs sensitively detect the early cancer stages than conventional methods (lines 112 & 128)? It could be explained with an example and clarify the possible question.

10.  In lines 131-132, the sentence could be more precise. For example, what is the “type of nanoparticles”?

11.  In lines 133-133, What nanocomposite could accurately penetrate the cancer cells (what cancers)? What is its penetration depth? For example, can they penetrate brain cancers by crossing the BBB or the blood-brain tumor barrier?

12.  I don’t think section 4 (line 143) is necessary to discuss the synthesis techniques of NPs. However, if authors wish to include it, the synthesis of NPs could narrow down individually (according to the graphical abstract).

13.  Does photosensitizers are not counted as nano-theranostic agents? Lines 159-160.

14.  The sentence requires a reference “To elucidate, NPs have been applied to image, screen and early diagnosis of the CRC (colorectal cancer) patients.” Lines 160-161.

15.  In line 168, how does the NPs are used to detect biomarkers? Again, it must be elaborate and compared with the traditional detection method.

16.  How does figure 3 refers to the sentence in lines 173-174? It’s not conclusive. Why in figure 3 has a thermometer image? What does it say to figure? MRI images were acquired at room temperature. Figure 3 is not providing any information. When discussing MRI contrast images, it would be better to provide one with magnetic nanoparticles MRI contrast images or suitable NPs for photoacoustic images. https://doi.org/10.7150/thno.4006, http://dx.doi.org/10.1166/jbn.2018.2546, https://doi.org/10.1021/acsomega.8b03283.

17.  References are required for each sentence in lines 183-187.

18.  Table 1. noticed typos and should be fixed accordingly. Under the column of types of nanoparticles, the semiconductor nanoparticles are not in the scope of the current study and could be removed. For ceramic nanoparticles, examples of what kind of imaging is used for cancer diagnosis should be mentioned.

19.  The title of section 6 could be rewritten precisely. (Line 203)

20.  The sentence in lines 216-218 required a reference.

21.  In line 227, what are several types of NPs according to its property? And the following sentence, what are the similar studies addressing? Both sentences are unclear.

22.  In line 229, what are up-conversion NPs? Does it necessary to discuss this here?

23.  The sentence needs to be clarified in line 240. The following sentence in line 241 is outside the scope of this study.

24.  The authors must first discuss the importance of functionalizing polyethylene glycol (PEG) to NPs, which could help in long circulations, before the sentences in lines 242-246.

25.  Table.2 typos need to be fixed accordingly. For the column of Nano type, it should maintain a consistent format: E.g., Post-treatment of Platinum nanoparticles (Pt NPs).

26.  In section 7.1.2. from lines, 274-280 require references for each sentence.

27.  Figure 4. Captions state various types of gold NPs. It looks like there are no various gold NPs.

28.  Tables 3 and 4 have typos that need to be fixed accordingly.

29.  In line 293, the proper usage of ‘nano magnetic particles’ should be ‘magnetic nanoparticles.

30.  One could include a reference for functionalized or hybrid magnetic nanoparticles to provide multitasking capabilities, including imaging, targeting, photothermal and hyperthermic agents in cancer treatments (https://doi.org/10.3390/mi13081279).  

31.  Table 5. It needs to be fixed for typo errors.

The content provided in section 7 and the following few sub-sections must precisely address the study’s scope. For example, the NPs in cancer diagnosis are not highlighted; instead, they provide only the cancer treatments, and, in some sub-sections, they are discussed in one or two sentences or never discussed.

-          The section 7 heading could be changed to commonly used NPs in cancer diagnosis and treatment.

-          Please provide a reference when stating or making any statements throughout the sections or subsections.

If the discussing pattern mimics the below highlighted critical points in sub-sections would be more precise for the readers.

-          What are these NPs, how do they synthesize, and their merits and demerits?

-          NPs in cancer diagnosis, the methods involved, and their types. Are these methods more feasible than traditional methods? How do these NPs make a path in cancer diagnosis, and what are their milestones and importance, etc.?

-          NPs in cancer treatment/therapy, etc.

32.  Section 7.3 requires an additional discussion for polymeric NPs within the scope of the study, such as discussing their application in cancer diagnosis.

33.  Section 7.4 second paragraph could be moved to 7.2. or sections 7.2 and 7.4 could be merged.

34.  Sections 7.5 to 7.8 still need to be discussed on cancer diagnosis.

35.  Section 7.9 Liposomes, while discussing what liposomes are and how they form, could be cited to this reference “https://doi.org/10.2174/1389203720666190521114936” and even could adopt an image to show the types of liposomes. In addition, this study, https://doi.org/10.3390/cancers13153690,” showed that liposomes could improve photosensitizers' stability from the liposomes bi-layer, which could help in multiple IVIS fluorescence imaging for brain cancer.

36.  There are two 7.11 sub-sections.

37.  Section 7.11 mechanism of drug release from nanomaterials. Unlike the NP's delivery route, no drug-release mechanisms are seen in this section. Is this a mistake in the title?

38.  Section 7.12 clinical application of nanomaterials in cancer treatment. It could include a table listing various types of NPs involved in clinical cancer trials with more references.   

39.  In line 631, the ‘some manufactured nanoparticles’ should not be addressed in a generalized way. Instead, it could be addressed with examples of NPs or by mentioning the actual type of NPs.

40.  From lines 634-653, it is exactly repeated in 672-681.

Author Response

The authors revised most of the sections and were satisfied with the content provided within the study's scope. However, the manuscript required further improvements. The consistency of nanoparticles (NPs) in the introduction and the following sections must be maintained, and one should avoid new nanoparticles which are not discussed in detail. In addition, some areas are highlighted outside this study’s scope.  The comments and suggestions are listed below.

The authors must address missing acronyms and typos throughout the manuscript.  

I listed a few typos along with others, 43, 70, 81, 97, 121, 127, 160, 168, 177, 189, 197, 227, 230, 234, 236, 237, 267, 335, 350, 352, 360, 365, 380, 406, 469, 625, 628, 648, 652, 659, 664, 676, 677, 678, etc. In addition, most of the tables have typos and should correct accordingly.  

  1. The sentence from lines 22-25 can be rewritten in the abstract according to the graphical abstract.  

Response: Thank you for the comment. Sentence is corrected

  1. The sentence from lines 28-30, what are many nanomaterials? This sentence could be rewritten.

Response: Thank you for the comment. Sentence is corrected

  1. The sentence from lines 38-42 is too big and could be split into two sentences.

Response: Thank you for the comment. Sentence is corrected

  1. What is “this technology” in line 87? Again, this sentence could be rewritten.

Response: Thank you for the comment. Sentence is corrected

  1. References or origin of the data source need to be included in lines 92-95.

Response: Thank you for the comment. Reference is added

  1. As mentioned, NPs could detect early-stage 0/1 cancers (line 112). Stating this type of contradictory statement leads to possible questions. What kind of NPs and targeting agents are used to detect cancer, and what type of cancers? Can these NPs cross the blood-brain barrier (BBB) to detect brain cancers?

Response: Thank you for the comment. Sentence is corrected

  1. The NP’s mono-nuclear phagocytosis or the reticule-endothelial system escape mechanism could have been highlighted in the introduction or before discussing NPs’ increased blood circulation in the body (lines 114 & 130).

Response: Thank you for the comment. NP’s mono-nuclear phagocytosis is discussed in the introduction section.

  1. In line 116, “Moreover, nanoparticles can bind with the tumor,” what type of NPs and how could these NPs bind to cancer or tumor cells? Are they using any targeting agents modified to NPs? These sentences could be corrected accordingly.

  1. Response: Thank you for the comment. Sentence is corrected

  1. How could the NPs sensitively detect the early cancer stages than conventional methods (lines 112 & 128)? It could be explained with an example and clarify the possible question.

Response: Thank you for the comment. Sentence is corrected

  1. In lines 131-132, the sentence could be more precise. For example, what is the “type of nanoparticles”?

Response: Thank you for the comment. Sentence is corrected

  1. In lines 133-133, What nanocomposite could accurately penetrate the cancer cells (what cancers)? What is its penetration depth? For example, can they penetrate brain cancers by crossing the BBB or the blood-brain tumor barrier?

Response: Thank you for the comment. There has been reports that nanocomposites like gold, platinum, silver copper and cobalt having more than type of nanoparticles are better candidate for the cancer detection and cancer imaging. The nanocomposites help to penetrate the breast and colon cancer cells with great accuracy and produce better signals which help to diagnose the cancer.

  1. I don’t think section 4 (line 143) is necessary to discuss the synthesis techniques of NPs. However, if authors wish to include it, the synthesis of NPs could narrow down individually (according to the graphical abstract).

  1. Response: Thank you for the comment. We have included this nanomaterials synthesis part as suggested by one of the reviewers.
  2. Does photosensitizers are not counted as nano-theranostic agents? Lines 159-160.

Response: Thank you for the comment. Sentence is corrected

  1. The sentence requires a reference “To elucidate, NPs have been applied to image, screen and early diagnosis of the CRC (colorectal cancer) patients.” Lines 160-161.

Response: Thank you for the comment. Reference is added

  1. In line 168, how does the NPs are used to detect biomarkers? Again, it must be elaborate and compared with the traditional detection method.

Response: Thank you for the comment. Sentence is corrected for better clarity

  1. How does figure 3 refers to the sentence in lines 173-174? It’s not conclusive. Why in figure 3 has a thermometer image? What does it say to figure? MRI images were acquired at room temperature. Figure 3 is not providing any information. When discussing MRI contrast images, it would be better to provide one with magnetic nanoparticles MRI contrast images or suitable NPs for photoacoustic images. https://doi.org/10.7150/thno.4006, http://dx.doi.org/10.1166/jbn.2018.2546, https://doi.org/10.1021/acsomega.8b03283.

Response: Thank you for the comment. Figure 3 has been redesigned and inserted appropriately

  1. References are required for each sentence in lines 183-187.

Response: Thank you for the comment. Reference is added

  1. Table 1. noticed typos and should be fixed accordingly. Under the column of types of nanoparticles, the semiconductor nanoparticles are not in the scope of the current study and could be removed. For ceramic nanoparticles, examples of what kind of imaging is used for cancer diagnosis should be mentioned.

Response: Thank you for the comment. Sentence is corrected and information is updated

  1. The title of section 6 could be rewritten precisely. (Line 203)

Response: Thank you for the comment. Sentence is corrected

  1. The sentence in lines 216-218 required a reference.

Response: Thank you for the comment. References are added

  1. In line 227, what are several types of NPs according to its property? And the following sentence, what are the similar studies addressing? Both sentences are unclear.

Response: Thank you for the comment. Sentence is corrected

  1. In line 229, what are up-conversion NPs? Does it necessary to discuss this here?

Response: Thank you for the comment. Sentence is corrected

  1. The sentence needs to be clarified in line 240. The following sentence in line 241 is outside the scope of this study.

  1. Response: Thank you for the comment. Sentence is corrected

  1. The authors must first discuss the importance of functionalizing polyethylene glycol (PEG) to NPs, which could help in long circulations, before the sentences in lines 242-246.

Response: Thank you for the comment. Sentence is corrected

  1. 2 typos need to be fixed accordingly. For the column of Nano type, it should maintain a consistent format: E.g., Post-treatment of Platinum nanoparticles (Pt NPs).

Response: Thank you for the comment. Table 2 is corrected

  1. In section 7.1.2. from lines, 274-280 require references for each sentence.

Response: Thank you for the comment. References are added

  1. Figure 4. Captions state various types of gold NPs. It looks like there are no various gold NPs.

Response: Thank you for the comment. The caption of Figure 4 is corrected

  1. Tables 3 and 4 have typos that need to be fixed accordingly.

Response: Thank you for the comment. Table 3 and Table 4 is corrected

  1. In line 293, the proper usage of ‘nano magnetic particles’ should be ‘magnetic nanoparticles.

Response: Thank you for the comment. Table 2 is corrected

  1. One could include a reference for functionalized or hybrid magnetic nanoparticles to provide multitasking capabilities, including imaging, targeting, photothermal and hyperthermic agents in cancer treatments (https://doi.org/10.3390/mi13081279).  

Response: Thank you for the comment. Information is added with addition of new reference.

  1. Table 5. It needs to be fixed for typo errors.

The content provided in section 7 and the following few sub-sections must precisely address the study’s scope. For example, the NPs in cancer diagnosis are not highlighted; instead, they provide only the cancer treatments, and, in some sub-sections, they are discussed in one or two sentences or never discussed.

-          The section 7 heading could be changed to commonly used NPs in cancer diagnosis and treatment.

-          Please provide a reference when stating or making any statements throughout the sections or subsections.

If the discussing pattern mimics the below highlighted critical points in sub-sections would be more precise for the readers.

-          What are these NPs, how do they synthesize, and their merits and demerits?

-          NPs in cancer diagnosis, the methods involved, and their types. Are these methods more feasible than traditional methods? How do these NPs make a path in cancer diagnosis, and what are their milestones and importance, etc.?

-          NPs in cancer treatment/therapy, etc.

Response: Thank you for the comments, information has been provided with new references and information has been updated. Table 5 has been corrected. 

  1. Section 7.3 requires an additional discussion for polymeric NPs within the scope of the study, such as discussing their application in cancer diagnosis.

Response: Thank you for the comment, information pertaining to cancer diagnosis is added

  1. Section 7.4 second paragraph could be moved to 7.2. or sections 7.2 and 7.4 could be merged.

Response: Thank you for the comment, information has been adjusted in the respective place.

  1. Sections 7.5 to 7.8 still need to be discussed on cancer diagnosis.

Response: Thank you for the comment, information pertaining to cancer diagnosis is added

  1. Section 7.9 Liposomes, while discussing what liposomes are and how they form, could be cited to this reference “https://doi.org/10.2174/1389203720666190521114936” and even could adopt an image to show the types of liposomes. In addition, this study, “https://doi.org/10.3390/cancers13153690,”showed that liposomes could improve photosensitizers' stability from the liposomes bi-layer, which could help in multiple IVIS fluorescence imaging for brain cancer.

Response: Thank you for the comment, references have been added

  1. There are two 7.11 sub-sections.

Response: Thank you for the comment, 7.11 section has been corrected

  1. Section 7.11 mechanism of drug release from nanomaterials. Unlike the NP's delivery route, no drug-release mechanisms are seen in this section. Is this a mistake in the title?

Response: Thank you for the comment, title has been modified to address the drug delivery of nanomaterials

  1. Section 7.12 clinical application of nanomaterials in cancer treatment. It could include a table listing various types of NPs involved in clinical cancer trials with more references.  

Response: Thank you for the comment, as there are only three studies were published regarding clinical trials, that’s the reason we have given described it briefly in the text form not in the table format

  1. In line 631, the ‘some manufactured nanoparticles’ should not be addressed in a generalized way. Instead, it could be addressed with examples of NPs or by mentioning the actual type of NPs.

Response: Thank you for the comment, the sentence is corrected

  1. From lines 634-653, it is exactly repeated in 672-681.

Response: Thank you for the comment, the sentence is corrected

Reviewer 3 Report

I am satisfied that all the queries are properly addressed by the auther. 

Author Response

I am satisfied that all the queries are properly addressed by the author.

Thank you so much

Reviewer 4 Report

In this manuscript, the authors describe that nanoparticles (NPs) can help in many cancer-related area. Some commonly used nanoparticles and their functions are well described. I find that the authors have put considerable effort into addressing the reports of the referees. They fix most of grammar errors. They add more references in “Polymeric-based nanoparticles”, “Magnetic nanoparticles” and “Dendrimers”. And they add descriptions about how combination of different NPs. But in part 7. Commonly used nanoparticles in cancer diagnosis, there are still many grammar errors. I suggest the authors double check it again. If those grammar errors can be fixed, I have no problem recommending it for publication.

Author Response

In this manuscript, the authors describe that nanoparticles (NPs) can help in many cancer-related area. Some commonly used nanoparticles and their functions are well described. I find that the authors have put considerable effort into addressing the reports of the referees. They fix most of grammar errors. They add more references in “Polymeric-based nanoparticles”, “Magnetic nanoparticles” and “Dendrimers”. And they add descriptions about how combination of different NPs. But in part 7. Commonly used nanoparticles in cancer diagnosis, there are still many grammar errors. I suggest the authors double check it again. If those grammar errors can be fixed, I have no problem recommending it for publication.

Response: Thank you for the comment. The grammatical errors have been corrected in the revised manuscript

Round 3

Reviewer 2 Report

The authors significantly improved the manuscript for the standard quality of the publication.